# EFFICIENT INDOOR RADIO MAP PREDICTION WITH IMPROVED TRANSFORMERS AND ACTIVE SAMPLING STRATEGIES

*Zhihao Zheng*[†]     *Limin Xiao*[‡]     *Ming Zhao*[‡]     *Yunzhou Li*[‡] [⋆]

[†] Department of Electronic Engineering, Tsinghua University
[‡] Beijing National Research Center for Information Science and Technology

## ABSTRACT

In this paper, we present a deep neural network designed for sampling-assisted pathloss radio map prediction, developed in the context of the MLSP 2025 "Sampling-Assisted Pathloss Radio Map Prediction Data Competition." The proposed model is built upon a U-Net encoder–decoder architecture and incorporates an enhanced Transformer module to strengthen global feature modeling capabilities. The network is trained on radio map data with sparsely sampled pathloss values. Experimental results show that our method achieves a weighted root mean square error (wRMSE) of 4.94 dB across both competition tasks, ranking third overall among all participating teams. These results highlight the model's strong prediction accuracy and generalization performance, particularly under sparse sampling conditions.

***Index Terms***— Wireless communications, Radio map prediction, Indoor pathloss, Channel model, Deep neural network

## 1. INTRODUCTION

In recent years, with the increasing complexity and diversity of wireless communication systems, there has been a growing interest in high-accuracy prediction of radio maps. A radio map describes the spatial distribution of wireless channel characteristics over a geographical area, and holds great potential for a variety of critical applications, such as high-precision localization [1], base station deployment optimization, UAV relay node positioning, and intelligent path planning [2]. However, due to complex environmental factors such as building obstructions, multipath effects, and dynamic scene changes, accurately modeling and efficiently predicting radio maps remains a highly challenging task.

### 1.1. Radio Map Prediction

Existing radio map prediction methods can generally be categorized into three types: deterministic models, empirical models, and data-driven models. Deterministic models rely on electromagnetic propagation theory, explicitly modeling physical effects such as scattering, reflection, and diffraction to calculate pathloss. For instance, ray tracing methods [3] can provide highly accurate predictions under ideal conditions but are computationally expensive and difficult to scale to large or real-time applications.

In contrast, empirical models [4, 5] are based on simplified assumptions and statistical correlations between pathloss and parameters such as distance, antenna height, and frequency. While they offer fast estimation, their lack of adaptability to complex and dynamic environments often results in significant prediction errors. Similarly, traditional data-driven approaches, such as the K-Nearest Neighbors (KNN) algorithm, also suffer from degraded interpolation accuracy under sparse sampling or complex terrain conditions.

With the advancement of deep learning, researchers have begun leveraging neural networks to model wireless propagation patterns. Studies have shown that convolutional neural networks (CNNs) and U-Net-based models [6] can achieve high prediction accuracy and computational efficiency when trained on sufficiently large datasets [7, 8]. Furthermore, recent work [9, 10] has explored integrating semantic environmental information—such as building layouts and elevation maps—into machine learning models for pointwise pathloss estimation, offering promising pathways toward high-resolution and robust radio map construction.

### 1.2. Scope of The Challenge

To advocate further research in this direction and facilitate fair comparisons in the development of DL-based radio propagation models in the less explored case of PL radio maps in indoor environments, the Sampling-Assisted Pathloss Radio Map Prediction Data Competition [11] was launched at MLSP 2025, aiming to encourage researchers to explore high-resolution indoor pathloss prediction using ground truth PL

_______________

This work was supported by the National Natural Science Foundation of China (NSFC) under Grants 62394294.
⋆ Corresponding Author

samples with varying sampling rates from the propagation environment. The competition is divided into two tasks:

- Task 1 evaluates model performance under randomly selected sampling points, assessing the predictive capability without sampling strategy optimization;

- Task 2 requires participants to design their own sampling strategies under a fixed sampling budget, investigating how to distribute samples optimally to improve overall prediction accuracy.

### 1.3. Our Contribution

In response to these tasks, this paper proposes a deep neural network architecture that integrates an enhanced Transformer module. Inspired by previous work [7, 12], the proposed model is based on the U-Net framework and incorporates attention mechanisms with stronger spatial modeling capabilities. The goal is to improve prediction accuracy with the assistance of sparsely sampled ground-truth data and to validate the generalizability of the model in complex radio propagation environments. The proposed method is systematically evaluated in the MLSP 2025 The Sampling-Assisted Pathloss Radio Map Prediction Competition [11]. The main contributions of this study are summarized as follows:

- We propose a U-Net-based model with enhanced Transformer modules for efficient sparse sampling-assisted prediction, demonstrating superior spatial modeling and generalization performance.

- We develop a spatial structure-aware sampling strategy that optimizes sampling point distribution under limited sample rates, yielding improved prediction accuracy.

- Our method achieves a weighted root mean square error (wRMSE) of 4.94 dB and ranks third in the competition, demonstrating strong effectiveness and robustness.

## 2. SYSTEM MODEL

### 2.1. Radio Map Prediction Model

To enable accurate and efficient radio map prediction under limited sampling conditions, we design a lightweight and high-performance deep learning architecture. The proposed network is composed of three core components: (1) a U-Net-style encoder–decoder backbone, (2) an optimized attention module, and (3) a modified feed-forward network (FFN) module of the Transformer.

**U-Net-Based Encoder–Decoder Framework.** U-Net was originally introduced for medical image segmentation [6]. Its typical encoder–decoder structure, combined with skip connections, enables effective retention of spatial detail while maintaining a compact, efficient, and trainable architecture. Due to these advantages, U-Net has been widely adopted in various computer vision tasks. Notably, [7] was the first to systematically adapt the U-Net framework for radio map prediction, modeling the spatial characteristics of pathloss and achieving significant performance improvements across multiple evaluation metrics. This work demonstrated the applicability of U-Net in wireless propagation modeling and has since served as a foundational benchmark in deep learning-based radio map prediction.

However, traditional U-Net relies heavily on convolutional neural networks (CNNs), whose inherently limited receptive fields constrain their ability to model long-range dependencies and complex spatial relationships. In contrast to CNN-based U-Net architectures with limited receptive fields, Transformer-based models can effectively capture global spatial dependencies [13], which is crucial for accurate radio map prediction in complex indoor environments.

To address this challenge, we enhance the U-Net structure by incorporating Transformer modules with attention mechanisms. Compared to pure CNNs, Transformers offer stronger global modeling capabilities and are more effective at capturing long-range dependencies and spatial context. By integrating Transformer blocks within the U-Net architecture, our model retains the high efficiency and hierarchical reconstruction capacity of U-Net, while significantly improving its ability to represent global propagation patterns. This enhancement ultimately leads to improved prediction accuracy and generalization performance in pathloss estimation tasks.

As shown in Fig. 2, the model takes as input a four-channel matrix:

$$\mathbf{X} = [\mathbf{R}, \mathbf{T}, \mathbf{D}, \mathbf{P}] \in \mathbb{R}^{H \times W \times 4}$$

where $\mathbf{R}$, $\mathbf{T}$, $\mathbf{D}$ represent the reflection coefficient, transmission coefficient, and distance from the transmitter to each point on the grid, respectively. $\mathbf{P}$ denotes the ground-truth pathloss values at sampled points, with other entries masked or set to zero. To ensure input consistency for neural network processing, all radio map images are resized to a standardized resolution of 256×256 pixels, which facilitates uniform batch training

The initial feature extraction is performed by a shallow convolutional module:

$$\mathbf{F}_0 = \phi_{\text{init}}(\mathbf{X}) \in \mathbb{R}^{H \times W \times C}$$

where $\phi_{\text{init}}(\cdot)$ denotes a set of convolutional layers, and $C$ is the number of output channels, which is set to 48 in our implementation.

The extracted feature $\mathbf{F}_0$ is passed through a Transformer-based multi-layer U-Net encoder, producing a latent representation: $\mathbf{F}_l \in \mathbb{R}^{\frac{H}{8} \times \frac{W}{8} \times 8C}$ The decoder then progressively upsamples and concatenates features with their encoder counterparts via skip connections to generate the final output: $\mathbf{F}_N \in$

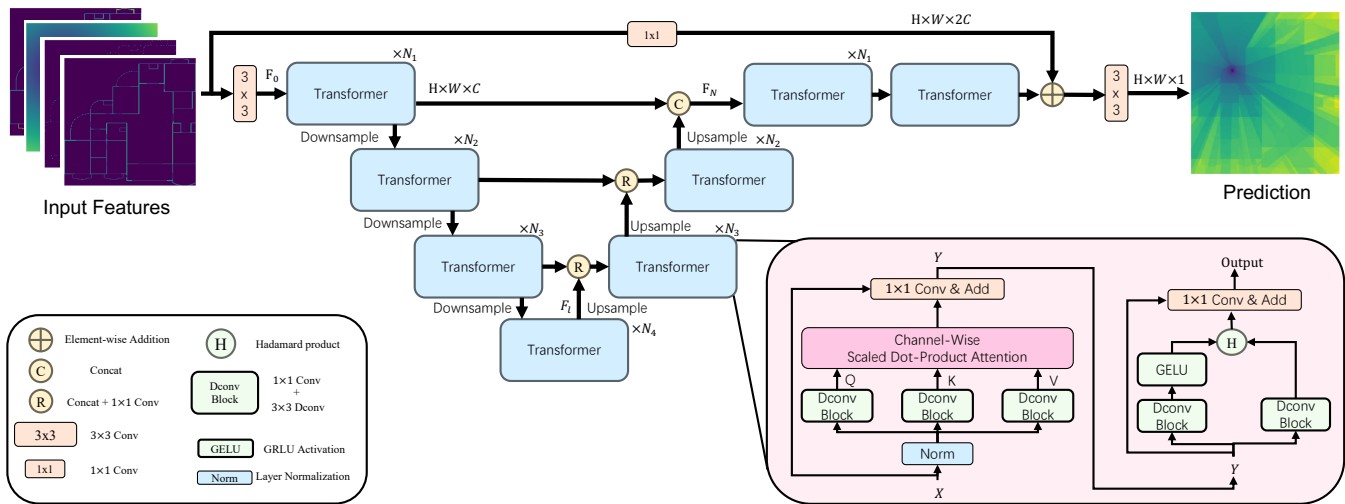

**Fig. 1**. Overview of the proposed model. The model integrates an encoder–decoder backbone with skip connections, an efficient attention module replacing standard self-attention with depthwise and $1 \times 1$ convolutions, and a gated feed-forward network for dynamic feature modulation. The input includes four-channel features, and the output is a high-resolution pathloss prediction map.



(a) Reflection coefficient  (b) Transmission coefficient  (c) Distance input  (d) Sample points

**Fig. 2**. Input Features. The four input channels are (a) the reflection and (b) transmission coefficients at each point of the grid, (c) the distance between Tx and each point, while the (d) is ground-truth pathloss values at sampled points.

$\mathbb{R}^{H \times W \times 2C}$. In this encoder–decoder architecture, the encoder reduces spatial resolution and increases feature dimensionality layer by layer to extract high-level semantic features. The decoder reverses this process via upsampling while reducing the channel dimension. Skip connections are established between each encoder and decoder layer to preserve fine-grained spatial details. Before concatenation, all skip connection features are processed with $1 \times 1$ convolutions to ensure channel alignment. This design mitigates information loss during fusion and enhances the model's ability to recover structural and boundary features.

**Optimized Attention Module.** In Transformer architectures, the self-attention mechanism is a powerful component for capturing global contextual dependencies. However, its computational complexity scales quadratically with respect to the input sequence length. Specifically, for a sequence of length $n$, the standard self-attention requires the computation of an $n \times n$ attention matrix, modeling interactions between all token pairs. This introduces significant computational and memory overhead, particularly in high-resolution inputs or long-sequence scenarios.

To mitigate this bottleneck, we adopt a structurally optimized attention mechanism inspired by the design in [12]. In our proposed design, the conventional fully connected linear layers in the self-attention module are replaced with a combination of lightweight $1 \times 1$ pointwise convolutions and depthwise separable convolutions. This modification reduces the number of parameters and computational cost, effectively lowering the attention complexity from $\mathcal{O}(n^2)$ to $\mathcal{O}(n)$.

As illustrated in Fig. 1, the optimized attention module maintains the ability to model long-range dependencies while significantly improving inference efficiency and scalability. Furthermore, the decreased memory footprint renders this architecture well-suited for deployment in resource-constrained environments, facilitating large-scale and real-time radio map prediction.

**Optimized Feed-Forward Network (FFN) Module.** To further enhance computational efficiency, we also redesign the Feed-Forward Network (FFN) module within the Transformer blocks. Instead of conventional fully connected layers, we employ a hybrid of $1 \times 1$ pointwise convolutions and depthwise convolutions. This configuration preserves expressive capacity while substantially reducing computational overhead.

Moreover, we incorporate a gating mechanism to dynamically regulate the flow of information. This gating structure enables the network to adaptively weight the features extracted from the attention module, thus allowing it to handle complex and diverse spatial feature distributions. The de-

tailed structure of this gated FFN module is presented in the pink box of Fig. 1.

Together, these architectural enhancements yield a more efficient and scalable Transformer framework, optimized for high-resolution spatial prediction tasks such as radio map estimation.

## 2.2. Sampling Point Selection Strategy

To ensure uniform spatial coverage when selecting sampling points from an input floor plan, we employ a stratified sampling strategy. In this approach, the floor plan is represented as a binary image, where pixels corresponding to free space (e.g., air) are labeled as 1, indicating valid sampling locations, and pixels corresponding to obstacles (e.g., walls, doors) are labeled as 0, indicating invalid sampling locations. The goal is to select $k$ sampling points that are evenly distributed across the accessible areas of the floor plan.

To achieve this, we divide the floor plan into a uniform grid of $s \times s$ cells, where $s = \lceil \sqrt{k} \rceil$. Within each cell, we randomly select one valid sampling point (i.e., a pixel labeled as 1). This process ensures that sampling points are spread across the entire floor plan, capturing spatial variations effectively.

However, due to the presence of obstacles, some grid cells may not contain any valid sampling points. To address this, we implement the following fallback strategies:

- **Case 1: Too many valid points.** If the number of selected sampling points exceeds $k$, we perform *random downsampling* to reduce the set to exactly $k$ points. This ensures uniform coverage across strata while maintaining randomness, and avoids bias toward large or dense cells.

- **Case 2: Too few valid points.** If fewer than $k$ points are selected—often due to some grid cells containing no valid pixels—we randomly sample additional points from the remaining *unselected but valid* pixels across the image until we reach $k$. This fallback preserves spatial diversity by drawing from under-sampled regions.

As shown in Fig. 3, consider a floor plan image of size $256 \times 256$ pixels with a sampling ratio $\rho = 0.0002$ (i.e., 0.02%). This results in $k = \lceil 256 \cdot 256 \cdot 0.0002 \rceil = 14$ sampling points. Accordingly, we set $s = \lceil \sqrt{14} \rceil = 4$, dividing the image into a $4 \times 4$ grid (each cell being $64 \times 64$ pixels). We then randomly select one valid sampling point from each of the 16 grid cells. If all cells contain valid points, we randomly downsample the 16 points to 14. If some cells lack valid points, we randomly select additional valid points from the remaining unselected valid pixels to ensure a total of 14 sampling points.

This stratified sampling approach ensures that the selected sampling points are uniformly distributed across the accessi-



**Fig. 3**. Sampling Point Selection Strategy

ble areas of the floor plan, effectively capturing the spatial structure and variations of the environment.

## 3. EXPERIMENTAL RESULTS AND ANALYSIS

### 3.1. Dataset

The dataset used in this challenge is based on [14], which contains indoor pathloss (PL) radio maps generated using Ranplan Wireless ray-tracing software. The competition use the Task 2 subset, consisting of 3750 radio maps from 25 indoor environments with diverse sizes, materials (e.g., concrete, drywall, wood, glass, metal), and three frequency bands (868 MHz, 1.8 GHz, 3.5 GHz, denoted as f1, f2, f3). The test set includes 200 maps from 5 unseen layouts under the 868 MHz band, each with 50 randomly placed transmitter locations.

### 3.2. Experimental Setup and Time Evaluation

All experiments were conducted using the PyTorch framework on an NVIDIA RTX 3090 GPU. For both tasks, the challenge dataset [14] was divided into training and validation sets based on indoor environments, with approximately 22 environments used for training and 3 reserved for validation. The detailed experimental configurations are summarized in Table 1.

It is worth noting that training was performed using data from all three center frequencies (868 MHz, 1.8 GHz, and 3.5 GHz) allowing the model to learn from a diverse set of propagation characteristics across different spectral bands. Although the test set includes only data at 868 MHz, this design choice was made to enhance the model's generalization ability and robustness to variations in frequency-dependent signal behavior.

**Table 1**. Experiment settings.

| Hyperparameter | Value |
| --- | --- |
| Learning rate | 3e-4 |
| Batch size | 1 |
| Optimizer | Adam |
| Maximum of epochs | 100 |
| Loss function | L1 Loss |
| $N_1, N_2, N_3, N_4$ | 4,6,6,8 |

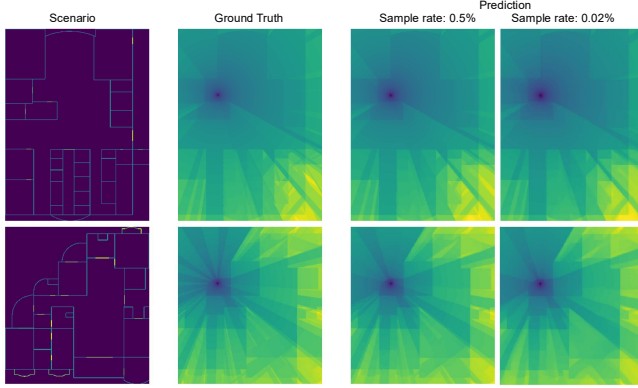

**Fig. 4**. Qualitative results on a validation set under selected samples. The first column shows the structural layouts of two indoor scenarios, followed by the ground-truth pathloss distribution. The third and fourth columns illustrate the model's predictions using 0.5% and 0.02% sampling rates, respectively.

The training time for both Task 1 and Task 2 was approximately 8 hours. During inference, we measured the time to process 100 test images, and found that on average, our model required approximately 135 ms per image, including data loading and prediction, on the RTX 3090 GPU.

### 3.3. Experimental Results

Fig. 4 presents illustrative qualitative results on the held-out validation set under selected samples. It can be observed that under both sampling conditions, the proposed model effectively reconstructs the global propagation structure, including signal decay and shadowing effects around obstacles. Notably, even at the extremely sparse 0.02% sampling rate, the model still maintains spatial coherence and captures key propagation patterns, albeit with slightly reduced detail compared to the 0.5% setting. This demonstrates the robustness and generalization ability of the proposed method under low-data regimes, and highlights the benefit of the designed sampling strategy in guiding effective pathloss estimation with minimal supervision. In addition, our method was evaluated by the competition organizers on a hidden test set comprising three representative indoor environments under a single frequency band. As reported in Table 2, the model achieved a third-place ranking among all participating teams, further validating its predictive accuracy and strong generalization performance.

As shown in Table 2, our proposed sampling strategy consistently outperformed the baseline random sampling method in Task 1, achieving an average improvement of approximately 2% across different sample rate. These results validate the effectiveness of our approach under various sampling conditions.

In addition, the competition organizers evaluated our

**Table 2**. Quantitative Results.

| Task | RMSE (dB) |
|---|---|
| Task1 (0.02%) | 6.36 |
| Task1 (0.50%) | 3.57 |
| Task2 (0.02%) | 6.27 |
| Task2 (0.50%) | 3.52 |
| Final (Weighted) | 4.94 |

**Table 3**. Comparison Under Different Sampling Rates.

| Sample Rate | RMSE (dB) |
|---|---|
| 0.00% | 7.00 |
| 0.02% | 6.27 |
| 0.50% | 3.52 |

model's performance on the test set both with selected sampling assistance and without any sampling guidance. As shown in Table 3, even without any sampling guidance, our Transformer-enhanced U-Net model achieved an RMSE of 7.00 dB, demonstrating its strong capacity to capture both global propagation trends and local detail. Introducing sampling guidance further boosted performance, yielding significantly better predictions across all evaluation metrics. Moreover, as the number of sampled points increased, the model's accuracy improved accordingly. These findings underscore that true sampling points, when used as supervision signals, can effectively guide the model in learning spatial pathloss patterns—enhancing its representational power and generalization in complex propagation environments. Thus, incorporating high-quality measurement samples, even in limited quantities, remains of practical and strategic value in real-world applications.

### 4. FUTURE WORK

Although the proposed method demonstrates promising experimental performance, there remain several areas for potential improvement. First, to enable batch training with uniform input dimensions, all radio map images were resized to a fixed resolution of $256 \times 256$. This resizing step may distort the underlying spatial characteristics of wireless signal propagation and hinder the model's ability to fully capture the physical laws of pathloss attenuation, thereby affecting prediction accuracy. In future work, we plan to explore position encoding and related techniques to mitigate the impact of resolution standardization and preserve spatial fidelity.

Second, in this study, the sampled pathloss values were directly fed into the neural network without additional processing. We believe that this approach does not fully exploit the spatial and contextual information embedded in the sampled points. To address this limitation, we intend to incorporate techniques such as spatial potential fields or loss fields [15] to better represent and utilize sampling information. This

could help the model more effectively learn propagation patterns and further enhance prediction performance.

Third, although our current sampling strategy already achieves a balanced spatial distribution, it does not yet fully adapt to heterogeneous room structures. In future work, we plan to further investigate room-aware sampling methods that dynamically adjust the sampling strategy according to individual room geometry, size, or material characteristics.

## 5. CONCLUSION

In this paper, we proposed a Transformer-enhanced model for indoor radio map prediction, developed in the context of the MLSP 2025 Sampling-Assisted Pathloss Radio Map Prediction Data Competition. Our model achieved an wRMSE of 4.94 dB on the hidden test set and ranked third overall among all participating teams. The experimental results and final ranking demonstrate that the proposed method is effective and well-suited for sampling-assisted pathloss radio map prediction tasks.

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
