# OpenReview forum: "Efficient Indoor Radio Map Prediction with Improved Transformers and Active Sampling Strategies"
_IEEE.org/MLSP/2025_SA_Radio_Map_Prediction_Challenge — SA Radio Map Prediction Challenge at MLSP 2025 Oral_

### Official Review · Reviewer_fQ9e · 2025-06-04
**This paper addresses the task of radio map prediction, proposing a novel deep learning architecture that integrates attention mechanisms within a U-Net framework. The authors aim to achieve accurate radio map predictions without relying on hand-engineered features.**

**Rating:** 7
**Confidence:** 5

**Review:**

Strengths:
1. The authors properly present the task of radio map prediction, the scope of the challenge and their contribution.
2. The proposed architecture represents a significant contribution, effectively combining the strengths of attention mechanisms with a U-Net-based structure for improved performance in radio map prediction.
3. A key strength of this work is the achievement of competitive performance without the necessity of manually engineered features, highlighting the efficacy of the end-to-end learning approach.
4. The "Future Work" section is well-articulated, identifying pertinent gaps and suggesting promising avenues for subsequent research.

Weaknesses:
1. "Related Work" section is missing.
2. The tokenization step for scaled dot-product attention is unclear. Is it performed across channels?
3. It is unclear how the skip connection from a 4-channel input is added to generate a single channel output.
4. The authors utilize data from Task 2, which incorporates three distinct carrier frequencies. However, the frequency information is not explicitly used as an input to the neural network.
5. The paper lacks a comprehensive description of the training/validation split strategy. Does validation contain unseen frequencies, unseen building layouts or unseen sampling positions?
6. The authors mention that the hidden test set consists of 5 building layouts, but it is actually 3.
7. While the authors claim that the designed sampling strategy offers significant benefits, the reported performance gains when compared to randomly selected sampling points appear to be negligible.

Comments and Suggestions for Improvement:
1. It would be insightful to investigate the performance of the proposed network when trained on manually extracted features.
2. The paper would benefit from a detailed discussion of the data augmentation techniques employed during training. This information is essential for replication and for understanding the robustness of the model.
3. The authors state that the computational complexity of the attention layer is reduced from O(n^2) to O(n). A thorough explanation of the methodology used to achieve this reduction, including any specific architectural modifications or algorithmic optimizations, should be provided.
4. Making the training codebase publicly accessible would greatly benefit the research community by facilitating reproducibility, further development, and comparative studies.

---

### Official Review · Reviewer_HAp3 · 2025-06-05
**The manuscript proposes an indoor radio map construction method based on an enhanced Transformer model that improves the construction accuracy by implementing the model's long-range modeling capability.**

**Rating:** 7
**Confidence:** 4

**Review:**

Overall, the paper is well organized. However, there are some minor technical issues that need to be addressed.

1.Check that the short lines in the last paragraph of the introduction make sense.
2.Please standardize the use of full names or abbreviations for figures or tables.For example, chapter 2.2, paragraph 4.
3.As I understand it, the number of samples is downsampled when there are all valid points in the sampling frame, then discarding those points can be discussed further. Also, what to do if there are more invalid points, making the number of sampled points not up to the mark?
4.Indeed, as the first paragraph of the future work chapter describes the drawbacks of the RESIZE operation, would it not be possible to consider mentioning that the manuscript uses this strategy in the writing of the earlier chapters?

---

### Official Review · Reviewer_DCQi · 2025-06-08
**This paper proposes an efficient prediction model that integrates an enhanced Transformer module with a U-Net-based encoder-decoder architecture, achieving a balance between local feature extraction and global spatial dependency modeling.**

**Rating:** 7
**Confidence:** 4

**Review:**

Main Contributions and Strengths:
1. Proposal of an Efficient Deep Learning Architecture:
An improved Transformer module is integrated into a U-Net-based encoder-decoder structure. This design achieves both robust local feature extraction and global spatial dependency modeling, enabling highly accurate path loss prediction even in complex indoor environments.
2. Improved Computational Efficiency and Scalability:
The conventional fully-connected layers in the self-attention mechanism are replaced with 1×1 and depthwise convolutions, reducing computational cost and memory consumption. This architecture ensures efficiency that is adaptable to high-resolution inputs and real-time inference.

Weaknesses and Areas for Improvement:
1. Lack of Ablation Studies:
More detailed ablation experiments are needed to convincingly demonstrate the contribution of each component to the final performance.
2. Discussion on the Impact of Spatial Resolution Standardization:
As mentioned by the authors in their Future Work, resizing inputs to 256×256 may distort physical distance information. Quantitative evaluation of this impact and possible solutions remain as future challenges.